# Liver X Receptor Agonist Inhibits Oxidized Low-Density Lipoprotein Induced Choroidal Neovascularization via the NF-κB Signaling Pathway

**DOI:** 10.3390/jcm12041674

**Published:** 2023-02-20

**Authors:** Tong Wu, Xinli Wei, Kuanrong Dang, Mengzhang Tao, Baozhen Lv, Tao Chen, Zuoming Zhang, Jian Zhou, Hongjun Du

**Affiliations:** 1Department of Ophthalmology, Eye Institute of Chinese PLA, Xijing Hospital, Fourth Military Medical University, Xi’an 710032, China; 2Department of Anatomy and Histology and Embryology, Xi’an Health School, Xi’an 710032, China; 3Center of Clinical Aerospace Medicine, Fourth Military Medical University, Xi’an 710032, China

**Keywords:** oxidized low-density lipoprotein, age-related macular degeneration, choroidal neovascularization, liver X receptor, NF-κB

## Abstract

Age-related macular degeneration (AMD) is the most common blindness-causing disease among the elderly. Under oxidative stress, low-density lipoprotein in the outer layer of the retina is easily converted into oxidized low-density lipoprotein (OxLDL), which promotes the development of choroidal neovascularization (CNV), the main pathological change in wet AMD. Liver X receptor (LXR), a ligand-activated nuclear transcription factor, regulates various processes related to CNV, including lipid metabolism, cholesterol transport, inflammation, and angiogenesis. In this study, we evaluated the effects of the LXR agonist TO901317 (TO) on CNV. Our results demonstrated that the TO could inhibit OxLDL-induced CNV in mice as well as inflammation and angiogenesis in vitro. Using siRNA transfection in cells and *Vldlr^−/−^* mice, we further confirmed the inhibitory effects of TO against the inflammatory response and oxidative stress. Mechanistically, the LXR agonist reduces the inflammatory response via the nuclear translocation of NF-κB p65 in the pathway for NF-κB activation and by enhancing ABCG1-dependent lipid transportation. Therefore, an LXR agonist is a promising therapeutic candidate for AMD, especially for wet AMD.

## 1. Introduction

Seniors are more likely to experience severe vision impairment due to age-related macular degeneration (AMD), with contemporary treatments offering limited relief [1]. AMD is a chronic, age-related disease and is associated with multiple underlying factors, including photooxidative stress, complement activation, and cellular senescence [2]. Oxidative stress in the retinal pigment epithelium (RPE) is an important contributor to AMD [3]. Photoreceptor outer segments are phagocytosed by RPE cells as part of maintaining homeostasis. [4]. As age increases, the phagocytic function of RPE cells decreases, and exfoliated outer segments accumulate under a layer of RPE and around Bruch’s membrane, resulting in its thickening and drusen formation [5,6]. Phospholipids in drusen are easily converted into oxidized phospholipids (OxPLs), leading to RPE cell apoptosis, inflammation, and choroidal neovascularization (CNV), which is considered the hallmark of exudative AMD (or wet AMD) [7,8]. OxPLs have been detected in CNV membranes from patients with AMD [9,10]. 

OxPLs form in many dyslipidemia and lipid metabolic diseases, including atherosclerotic lesions, and frequently mediate proinflammatory changes [11]. Oxidized low-density lipoprotein (OxLDL) is the main form of OxPLs. It induces long-term proinflammatory cytokine production (e.g., TNF-α, MCP-1, interleukin-6 (IL-6), IL-18, CD36, matrix metalloproteinase 9 (MMP9), and SR-A) and foam cell formation via the epigenetic reprogramming of monocytes [12]. Cancer research has revealed that the NF-κB signaling pathway was activated by OxLDL via LOX-1, with the vascular endothelial growth factor (VEGF) expression upregulation following [12]. With the intravitreal injection of OxLDL and laser induction, we have previously established a CNV model, identifying OxPLs as a pathogenic component of CNV [13]. Further in vivo and in vitro studies have shown that IL-6, IL-1β, VEGF, MMP9, and CC chemokine receptor 2 (CCR2) expression levels are increased in CNV. Therefore, it is critical to controlling abnormal lipid metabolism, reducing the OxLDL accumulation, and further reducing the inflammatory response. 

An integral part of the nuclear receptor superfamily, liver X receptors (LXRs) act as transcription factors activated by ligands [14]. It is crucial for the maintenance of cholesterol homeostasis throughout the body for LXRs, in which cholesterol is moved from peripheral cells (for example, macrophages) to liver cells, called reverse cholesterol transport [15,16,17]. The LXR agonist TO901317 (TO) can activate LXRs and has been evaluated for the treatment of abnormal lipid metabolism and inflammation-related diseases, with positive results in atherosclerosis (AS) [18,19]. As OxLDL plays similar roles in the pathogenesis of AS and AMD, we hypothesized that LXR agonists can also inhibit CNV formation by reducing intracellular and subretinal lipid accumulation. The objective of this study was to provide a new understanding of wet AMD and its treatment by evaluating what effect TO has on OxLDL-induced CNV.

## 2. Materials and Methods

### 2.1. Animals

The Laboratory Animal Center of Fourth Military Medical University provided us with wild-type C57BL/6J mice of the male breed. Male B6.Vldlr knockout (*Vldlr*^−/−^) B6; 129S7-*Vldlr^tm1Her^*/J mice (Stock No. #002529; Jackson Laboratory, Bar Harbor, ME, USA) [20] were a gift from Pro. Zuo-Ming Zhang (Center of Clinical Aerospace Medicine, Fourth Military Medical University).

Six to eight week old mice were housed under a 12 h light/dark cycle with free availability of food and water. Veterinary ethics committees at the Fourth Military University reviewed the procedures for induced CNV by laser photocoagulation in mice, and a compliance with the ARVO Statement on Animal use in Ophthalmic and Vision Research was achieved.

### 2.2. Cell Culture

The Chinese Academy of Science provided ARPE-19 and RF/6A cells. They were routinely cultured in Invitrogen DMEM/F-12 (Carlsbad, CA, USA), which contains 10% fetal bovine serum (FBS), 100 units of penicillin per milliliter, and 100 units of streptomycin per milliliter at 37 °C in a humidified atmosphere with 5% CO_2_. 

### 2.3. Induction of Choroidal Neovascularization 

The laser photoagglutination-induced CNV model was performed in accordance with previous reports [21]. Sodium pentobarbital (1%, Sigma, St. Louis, MO, USA; P3761) was first administered intraperitoneally to male C57BL/6J mice to anesthetize them; compound tropicamide eyedrops (Santen Pharmaceutical, Osaka, Japan) and oxybuprocaine hydrochloride eyedrops (0.4%, Santen Pharmaceutical) were then applied to the eyes. An incident 532 nm laser (Twin, Quantel, France) was shone into the right eye of each animal while their pupils were dilated, inducing burns 1.5–2 disc diameters away from the optic nerve (75 μm, 100 ms, and 120 mV). As long as a vaporization bubble did not form with hemorrhage, laser spots that ruptured Bruch’s membrane were considered valid, and the CNV lesions were employed for analysis. Immediately after laser photocoagulation, animals were randomly divided into four groups.

### 2.4. Severity Assessment of CNV

An analysis of choroidal flat-mounts was performed seven days after laser photocoagulation to determine the size of the CNV. Briefly, in anesthetic mice, 0.9% saline solution was transcardially administered followed by 4% paraformaldehyde, and the entire ocular globe was removed. During the procedure, the anterior segment of the eye and the crystalline lens were surgically removed, as well as the retinas from the optic nerve head that were detached and separated. The remaining eye cups were flat-mounted using 4~6 incisions, after which the flat-mount formulations were permeabilized in a 0.2% Triton X-100 solvent for 24 h. Next, primary antibodies against CD31 (1:150; Abcam, Cambridge, UK) and F4/80 (1:150; Abcam, Cambridge, UK) were washed in cold PBS after being incubated overnight at 4 °C. Samples were incubated with a 1:200 dilution of secondary antibodies conjugated to fluorescence (Abcam, Abcam, Cambridge, UK) for two hours and then washed with cold PBS three times. With a fluorescence microscope system (BX51; Olympus, Tokyo, Japan), images of the flat mounts were acquired and CNV areas computed. 

### 2.5. Frozen Section Preparation and Immunofluorescence Staining

Perfusion and fixation were performed on the eyeball as described previously. Cryosections of 10 μm thickness were prepared by soaking the samples in TissueTek optimal cleaving temperature compound (Sakura Finetek, Torrance, CA, USA) following placement in a 30% sucrose resolution at 4 °C overnight. 

After being desiccated at 37 °C for 2 h, frozen eyeball sections were cleansed with PBS. It blocked by PBS comprising 1% BSA and 0.5% Triton X-100 for 2 h at room temperature. Slides were incubated with primary antibodies against CD31 (1:150; Abcam, Cambridge, UK), F4/80 (1:150; Abcam, Cambridge, UK), T15 (1:100; Avanti Polar Lipids, Alabaster, AL, USA), Ly6c (1:100; themo Fisher Scientific, Waltham, MA, USA), NF-κB (1:100; CST, Danvers, MA, USA) overnight at 4 °C and then secondary antibodies were applied for 2 h at room temperature, followed by counterstaining using DAPI for 15 min under the same conditions.

### 2.6. Cytoplasmic and Nuclear Protein Extraction and Western Blotting

Tissue or cell lysates were obtained in RIPA (Beyotime, Shanghai, China) by using proteinase and phosphatase inhibitors (Roche Molecular Biochemicals, Indianapolis, IN, USA). A Cytoplasmic Nuclear Separation Kit from Beyotime was used to extract proteins in nucleus and cytoplasm. The protein density was estimated using BCA protein assay reagent (Beyotime, Shanghai, China). An SDS-PAGE was used to separate proteins, followed by a PVDF membrane (Millipore, Bedford, MA, USA) transfer. After blocking the membranes for 2  h in 5% skim milk at room temperature, they were incubated with primary antibodies under 4 °C overnight. Next, the membranes were incubated with secondary antibodies for 1 h. Afterwards, secondary antibodies were applied to the membranes for one hour. Following three washes with TBST, intensive chemiluminescence systems (Millipore, Bedford, MA, USA) were used to measure the protein’s signals and standardized against β-actin, GAPDH or Histon H3 levels. The antibodies are listed in Appendix A.

### 2.7. Real-Time Quantification of Polymerase Chain Reactions

RNA in total of choroidal tissue or ARPE-19 cells was isolated through the use of the TRIzol Purification Kit (Thermo Fisher Scientific, Waltham, MA, USA). An RT (reverse transcription) system (TaKaRa Dalian, Dalian, China) was applied to create the cDNA templates. Polymerase chain reaction (PCR) was conducted in triplicate utilizing a kit (SYBR Premix EX Taq; TaKaRa Dalian, Dalian, China) and the ABI Step One Plus Real-time PCR System, with β-actin serving as an in-house control. A list of the PCR primers can be found in Appendix A.

### 2.8. Tube Formation

A 48-well plate lubricated with Matrigel was seeded with 50,000 RF/6A cells per well and grown in DMEM incorporating OxLDL or TO in a humidified incubator at 37 °C in 5% CO_2_. Each well was randomly inspected with an inverted microscope following treatment to examine tube formation at indicated time points. The total tube length of the tubular structure was counted with ImageJ 2.0, and the outcomes are presented as the mean multiple change compared with the pipe length in the controlling group. A minimum of three wells per situation were seeded, and the experiments were replicated three times each.

### 2.9. Oil Red O Staining

Frozen sections of eyeballs were allowed to dry at room temperature for 2 h and subsequently washed with PBS. Then, sections were rinsed with 60% isopropyl alcohol for 15 s to promote neutral fat staining. Oil red O stock solution was mixed with deionized water at a ratio of 3:2, placed at room temperature with 10 min, and then filtered through a 0.22 μm capillary membrane. After 15 min of soaking in strained Oil Red O working solution in dark conditions, the tissue was rinsed. A solution of 60% isopropanol was applied to the samples for 15 s, followed by three washes with PBS. After the tissues were counterstained with hematoxylin, they were mounted, and images were obtained under a microscope.

### 2.10. RNAi

One day before transfection, 3–4 × 10^5^ cells were added to a 6-well culture plate and cultured with antibiotic-free medium until the cell density reached 60%. Then, 5 μL of Lipofectamine2000 and siRNA storage solution diluted in 200 μL of serum-free culture medium Opti-MEM were incubated for up to 5 min at room temperature. The liquids were gently mixed to form a Lipo2000-siRNA mixture, and cells were added after standing at room temperature for 20 min. The complete medium containing serum was replaced after 4–6 h, followed by fluorescence microscopy check the transfection efficiency after 18 h of transfection.

### 2.11. Protein Array Analysis

The Proteome Profiler Array (Human Angiogenesis Kit; R & D Systems Europe, Abingdon UK) was used to recognize site-specific pro-angiogenic elements in the condensed culture media. Cell protein lysates were obtained and detected using the kit. Arrays were prepared and incubated according to manufacturer recommendations. A 30-min incubation at room temperature was performed with the streptavidin-HRP solvent. Membranes were laundered three times, and signals were detected by gel chemiluminescence imaging. ImageJ was used to quantify the pixel intensity in each spot after scanning, adjusting size, and inverting the array films using Adobe Photoshop. After calculation of the mean signal of duplicate spots corresponding to each protein, a clear region of the membrane was subtracted from the background signal.

### 2.12. Data Analysis

SPSS 22.0 was used to conduct the statistical analyses. The data are expressed as means and standard deviations (SD). An analysis of variance using Student’s t-tests was performed between the two groups. Comparisons were carried out between the two groups by means of Student’s *t*-tests. An analysis of variance (ANOVA) and a least significant difference (LSD) test was performed to contradistinguish differences among multiple groups. Categorical data were compared using the chi-square test. Double-tailed P-values of minus 0.05 indicated significance.

## 3. Results

### 3.1. LXR Agonist Inhibits CNV Formation and Inflammation

Consistent with previous research, we found that OxLDL stimulates oxidative damage in AMD. Accordingly, we investigated how LXR affects the occurrence of CNV. A mouse model of laser-induced CNV was treated with 25 μg/mL OxLDL and 10 μM TO (final concentration) by vitreous injection. The CNV field was statistically smaller in the OxLDL + TO cohort than in the OxLDL cohort (Figure 1a,c), indicating that the LXR agonist inhibited CNV formation. Lipid accumulation (as evaluated by T15) and macrophages activation (marked by F4/80 and LY6C) were reduced in the CNV area after 10 μM TO injection (Figure 1b and Appendix A). qRT-PCR with choroidal tissues showed that levels of IL-6, CCR2, and VEGF were lower in the OxLDL+TO group than in the OxLDL group (Figure 1d). 

### 3.2. LXR Agonist Inhibits Angiogenesis in RPE Cells

To further establish the effects of LXR, we cultured RPE cell lines, which play a role in phagocytosis and inflammatory release during CNV formation. After treatment with 25 μg/mL OxLDL and 10 μM TO, cell secretions containing cytokines were added to retinal/choroidal vascular endothelial cells (RF/6A) to simulate neovascularization in vitro. An increased tube formation showed by tube length, number of tube loops, and tube branch was observed in the OxLDL group and a reduced tube formation was observed in the OxLDL + TO group. These results demonstrated that the inhibitory effects of TO on angiogenesis can be observed in vitro (Figure 2a,b). Cell secretions were used to detect angiogenesis-related protein expression with a protein array, revealing that TO reduces the levels of some angiogenesis-promoting cytokines Coagulation Factor III (TF), DPPIV, Endoglin, IGFBP3, Serpin E1 (PAI-1), Thrombospondin-1, and VEGF were secreted by RPE cells (Figure 2c,d). 

### 3.3. LXR Agonist Inhibits Lipid Accumulation via the ABCG1 Pathway and Inflammation in RPE Cells

To determine the specific mechanism underlying the effect of LXR, the phagocytosis and inflammation release functions of RPE cells were considered. Dil-OxLDL was phagocytosed by RPE cells, with 10 μM TO reduced (Figure 3a,b) and this phagocytosis via the lipoprotein regulatory pathway ABCG1 (Figure 3c). In vitro, as determined by western blotting, various inflammatory factors involved in angiogenesis, including CCR2, TNF-α, IL-1β, IL-6, and VEGF, were expressed more highly in the OxLDL group than in the control group. IL-1β and VEGF were attenuated by TO. Levels of the chemokine CCR2 and inflammatory factor IL-6 did not differ among these groups (Figure 3d,e). 

### 3.4. LXR Agonist Inhibits the OxLDL-Activated NF-κB Pathway

Interleukin-1 (IL-1) family cytokines bind to cognate receptors, initiating an NF-κB signal transduction cascade intracellularly. Both innate and adaptive autoimmune systems use it as a pivotal signaling molecule to cause inflammation when a variety of stimuli are encountered [22]. In vivo protein expression analyses further revealed that 25 µg/mL OxLDL activated AKT and NF-κB signaling pathways at 5 days, and 10 μM TO inhibited this process (Figure 4a,b). In vitro, as the TO concentration increased, the expression of inflammatory factors by RPE cells induced by 25 µg/mL OxLDL decreased. OxLDL was upregulated in coordination with IL-1β and TNF-α by activating the NF-kB pathway, and TO can inhibit this process (Figure 4c,d).

### 3.5. LXR Knockout Aggravated NF-κB Pathway-Mediated IL-1β and VEGF Expression

RNAi technology was used to knock down LXR in cells, with an efficiency of almost 50% (Figure 5a,b). Western blotting showed that LXR knockout generally increased the expression of NF-κB p-p65, IL-1β, and VEGF, demonstrating that the NF-κB pathway was synergistically up-regulated, further supporting the inhibitory effect of LXR (Figure 5c,d).

### 3.6. LXR Agonist Inhibits OxLDL-Mediated Activation of NF-κB p65 Nuclear Translocation

Activation of NF-κB pathway involves the transcriptional activation of target molecules after the dimerization of p65 entry in the nucleus (Figure 5e,f). A cytoplasmic proteins analysis showed that IκB levels were higher in OxLDL compared with the control group, and they were reduced in the OxLDL + TO group. With respect to nuclear proteins, NF-κB p65 levels were increased in the OxLDL group and reduced in the OxLDL + TO group (Figure 5g). These results confirmed that p65 partially enters the nucleus after OxLDL treatment and TO can rescue the process, providing insight into the specific molecular mechanism by which LXR inhibits OxLDL-activation of the NF-κB p65 pathway.

### 3.7. LXR Agonist Reduces the CNV Volume in Vldlr^−/−^ Mice

To evaluate whether the OxLDL-induced CNV can be inhibited by TO in vivo, *Vldlr^−/−^* mice fed a high-fat diet were selected as an animal model. *Vldlr^−/−^* mice showed higher cholesterol and triglyceride levels than WT mice (Figure 6a). In the laser-induced CNV model of *Vldlr^−/−^* mice, immunofluorescence staining of frozen sections showed that the intravitreal injection of 10 μM TO reduces the area of CNV formation. In addition, Oil red O staining lipid accumulation in the CNV area decreased in response to TO, indicating that the RPE cell phagocytosis of OxLDL is reduced (Figure 6b). The CNV region was larger in the *Vldlr^−/−^* mouse model than in the control group, and the increase was attenuated by TO injection (Figure 6c). 

## 4. Discussion

Lipid accumulation and inflammation are crucial components of the pathogenesis of wet AMD [23,24]. Therefore, the process of AMD could be decelerated by the inhibition of these processes. This study provides experimental evidence for the inhibitory effect of LXR against the inflammatory response and lipid accumulation via inhibition of the NF-κB signaling pathway in RPE cells and in an animal model of advanced CNV, supporting the therapeutic effects of LXR agonist TO in AMD and providing insight into the mechanisms underlying these effects.

The hallmark of exudative AMD is CNV through Bruch’s membrane, with intraretinal or subretinal leakage, hemorrhage, and RPE detachment [25]. CNV develops as a result of extensive drusen and pigmentary irregularities caused by lipid and protein deposition of the retinal pigment epithelium, with inflammation [26,27]. Drusen formation is very similar to the formation of atherosclerotic plaques, i.e., a large number of lipids in the Bruch membrane are oxidized to OxPLs under high oxidative stress [28]. Drusen forms under the phagocytic lipid function of RPE cells, similar to macrophages, which can induce the expression of adhesion molecules, chemokines, and proinflammatory cytokines, and then develop into CNV under an inflammatory environment. In the early stage, we established an advanced CNV mouse model of oxidative stress and lipid accumulation [13]. Based on this model, we explored the reduction of OxLDL accumulation by lipid metabolism, which may fundamentally reduce inflammation and neovascularization.

As a candidate therapeutic target, LXR (nuclear receptor subfamily 1 group H, NR1H) regulates cholesterol homeostasis and inflammation, two pathways affecting cholesterol levels [29]. Increasing age can cause decreased LXR expression inside the RPE, and mice without LXR developed ocular pathologies dependent on their isoform [30]. A mouse model of ocular inflammation showed that LXR activation improved lipid accumulation and oxidant-induced injury in RPE cells as well as decreased lipid deposition in ocular tissues. [30,31]. We obtained similar results in our animal model. In addition, we explored the in-depth mechanisms in cell experiments. Further arguments for research into the involvement of LXR in eye aging derives from genome-wide association surveys that have revealed a linkage between the genetic association of single nucleotide polymorphisms implicated in cholesterol metabolic transport and AMD, including human transporter subfamily ATP binding cassette transporters [32]. In this study, the LXR agonist (TO901317) reduced lipid accumulation as well as CNV formation induced by the inflammatory response. Cholesterol metabolism transport in RPE cells is based on ABCG1 instead of ABCA1, and the inflammatory response depended on the NF-κB signaling pathway.

There is increasing evidence that NF-κB pathway activation is associated with the progression of CNV formation [33]. NF-κB is an important nuclear transcription factor [34]. It combines with its natural inhibitory protein IκB to form a dimer. IκB kinase is activated by its protein phosphorylation or ubiquitination, and the IκB protein is degraded. NF-κB p65 is released and migrates into the nucleus, where it acts as a transcription factor [34,35]. Our results demonstrated that IκBα activation promotes the nuclear translocation of the p65. A proinflammatory cytokine production pathway is also activated by NF-B activation, which could result in angiogenesis. A proposed therapeutic strategy for AMD involves suppressing the NF-B pathway. A recent report by Biswas et al. [23] has disclosed that inhibition of the NF-κB pathway can reduce the inflammatory response in RPE cells. Our study demonstrated that TO901317 suppressed the nuclear translocation of p65 and reduced inflammation.

## 5. Conclusions

In conclusion, our results revealed that an LXR agonist (TO901317) could protect RPE cells against OxLDL stimulation by inhibiting the inflammatory response and oxidative stress. Furthermore, the protective role of TO901317 on CNV formation was mediated by suppression of the NF-κB signaling pathway, and its transport cholesterol metabolism in RPE cells on ABCG1. Therefore, an LXR agonist is a potential therapeutic agent to delay the progression of AMD.

## Figures and Tables

**Figure 1 jcm-12-01674-f001:**
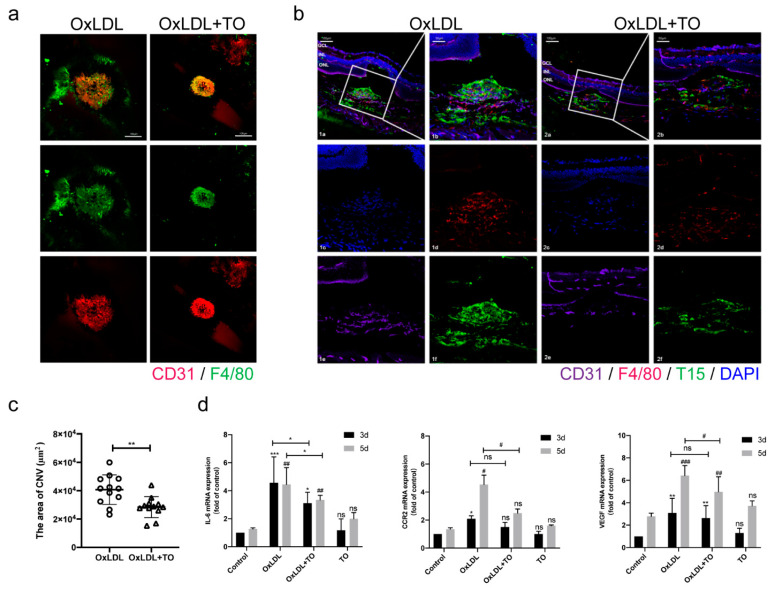
Effect of LXR agonist on OxLDL-induced experimental CNV. (**a**) Choroidal flat mount with immunofluorescence double labeling staining, CD31 labeled endothelial cell (red), F4/80 labeled macrophage (green), and DAPI labeled nuclear (blue), bars = 100 μm (**b**) Frozen section with immunofluorescence staining, CD31 labeled endothelial cell (purple), F4/80 labeled macrophage (red), T15-labeled LDL (green), and DAPI labeled nuclear (blue), bars = 100 μm (**c**) Quantitative analysis of 2D lesion regions in CNV mice post laser processing in choroidal flat mounts. The lesion sizes were gauged with the ImageJ software. Values are presented by means ± SEM, n = 12, ** p* < 0.05, *** p* < 0.01 vs. OxLDL group (**d**) IL-6, CCR2, and VEGF mRNA in choroidal tissue expression were detected via qRT-PCR. Data are expressed as the relative CNV lesion area ± SEM, n = 3, ns indicates *p* > 0.05, * *p* < 0.05, ** *p* < 0.01, *** *p* < 0.001 vs. 3 day control group, # *p* < 0.05, ## *p* < 0.01, ### *p* < 0.001 vs. 5 day control group.

**Figure 2 jcm-12-01674-f002:**
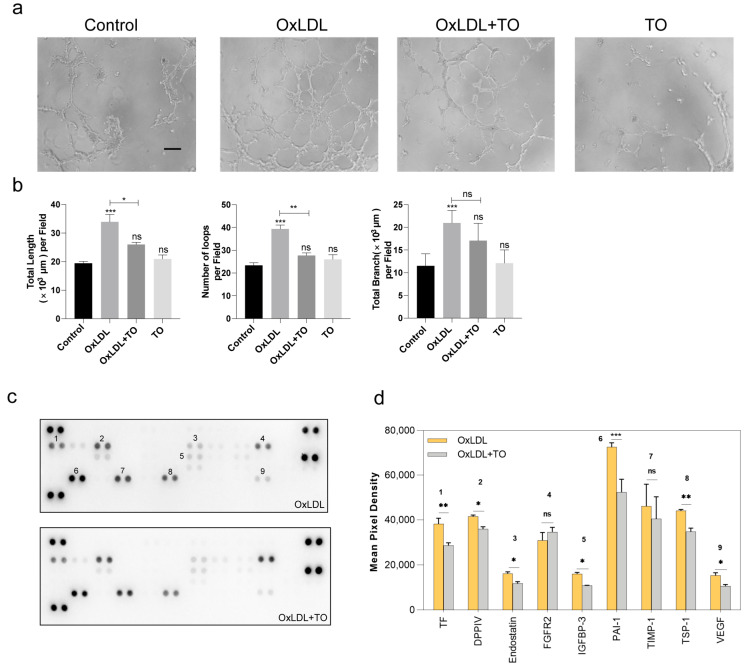
Effect of LXR agonist on OxLDL pretreatment to RPE cell. (**a**) RF/6A cell tube formation through RPE cell cytokines treatment. (**b**) The quantitative results of tube formation. Evaluation of tube formation by the number of loops, the total length, and branch length of tube in RF/6A cell. bars = 500 μm. Values are expressed as means ± SEM, ns indicates *p* > 0.05, ** p* < 0.05, *** p* < 0.01, **** p* < 0.001 vs. control group (**c**) The Human Angiogenesis Array detects multiple analytes in cell culture lysates. (**d**) Histogram profiles of specific analytes are created by use of imagery software to quantify the average speckle pixel density from the array membrane. Values are expressed as means SEM, ns indicates *p* > 0.05, * *p* < 0.05, ** *p* < 0.01, *** *p* < 0.001 vs. OxLDL group, 1 indicates TF, 2 indicates DDPIV, 3 indicates Endostain, 4 indicates FGFR2, 5 indicates IGFBP-3, 6 indicates PAI-1, 7 indicates TIMP-1, 8 indicates TSP-1, 9 indicates VEGF.

**Figure 3 jcm-12-01674-f003:**
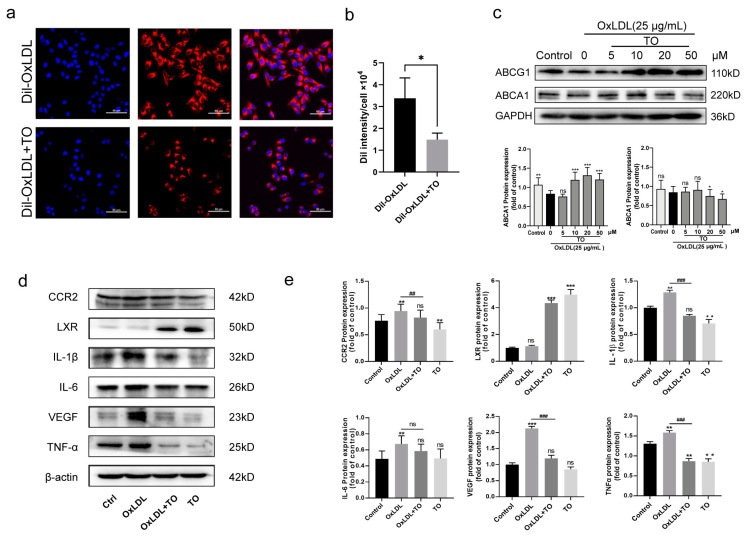
Lipid accumulation and inflammation effect of LXR agonist on OxLDL pretreatment to RPE cell. (**a**) RPE cells were incubated with 25 μg/mL OxLDL and 10 μM TO as indicated for 12 h in culture. The phagocytosis of Dil-labeled OxLDL examined by fluorescence staining, Dil-labeled OxLDL (red) and DAPI labeled nuclear (blue), bars = 20μm. (**b**) The quantitative results of Dil-OxLDL intensities. Data shown are the mean ± SEM, n = 3, * *p* < 0.05 vs Dil-OxLDL group. (**c**) To detect and quantify ABCG1 and ABCA1, Western blotting was applied. Values are expressed in terms of means ± SEM, ns indicates *p* > 0.05, ** p* < 0.05, *** p* < 0.01, **** p* < 0.001, * vs. control group (**d**) Western blotting was used to detect the expression of inflammation factors. (**e**) Quantification of inflammation factors expression. Data are expressed as mean ± SEM, n = 3, ns indicates *p* > 0.05, * *p* < 0.05, ** *p* < 0.01, *** *p* < 0.001 vs. control group, ## *p* < 0.01, ### *p* < 0.001 vs. OxLDL group.

**Figure 4 jcm-12-01674-f004:**
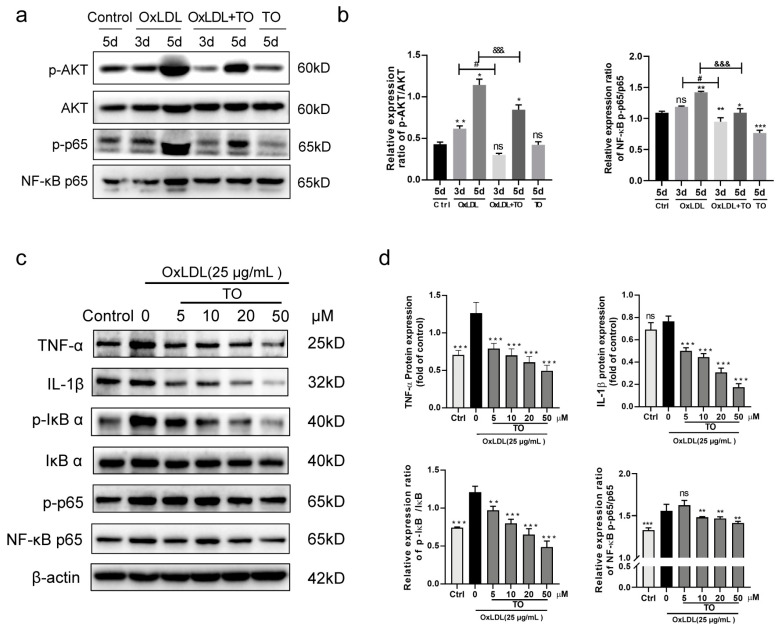
LXR agonists Inhibits OxLDL-activated the AKT and NF-κB pathway. (**a**) Using Western blotting, the presence of AKT and the NF-κB route in choroidal tissues of mice was determined. (**b**) Quantification of AKT and NF-κB pathway protein expression. n = 3, ns indicates *p* > 0.05, * *p* < 0.05, ** *p* < 0.01, *** *p* < 0.001 vs. 5 d control group, # *p* < 0.05 vs. 3 d OxLDL group, &&& *p* < 0.001 vs. 5 d OxLDL group (**c**) Western blotting was employed to detect the manifestation of NF-κB route in model RPE cell. (**d**) Quantification of TNF-α, IL-1β, and NF-κB pathway protein expression. Data are expressed as mean ± SEM, n = 3, ns means *p* > 0.05, * *p* < 0.05, ** *p* < 0.01, and *** *p* < 0.001 vs. OxLDL 25 μg/mL group.

**Figure 5 jcm-12-01674-f005:**
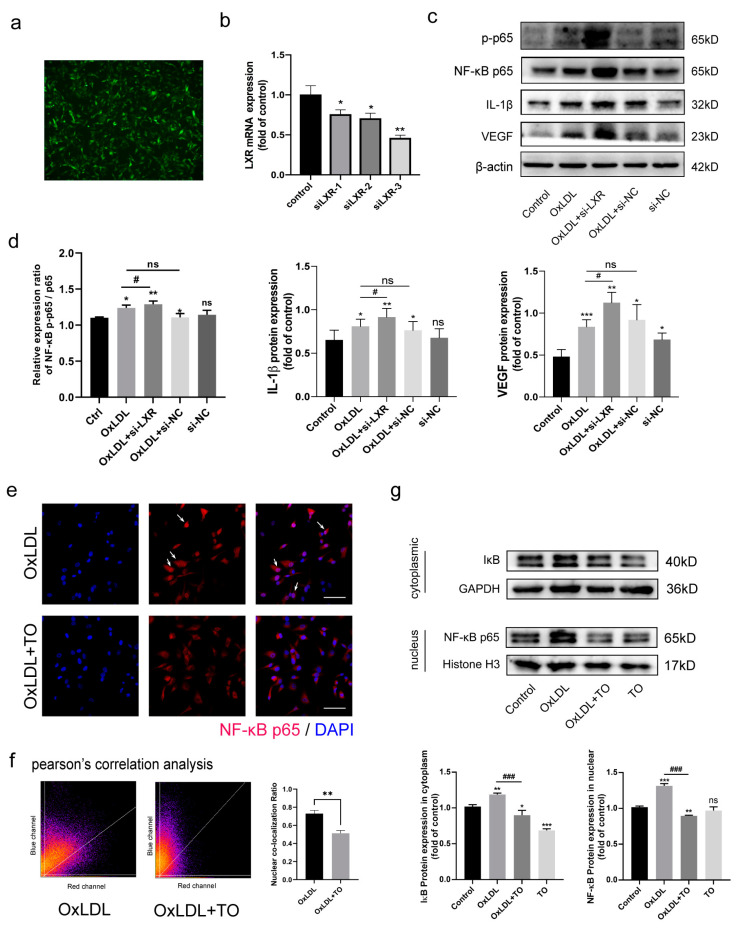
Effect of the specific LXR knockout and LXR agonists on the NF-κB route. (**a**) Cell transfer siLXR-GFP efficiency. (**b**) LXR mRNA expression, n = 3, * *p* < 0.05, ** *p* < 0.01 vs. control group. (**c**) Western blotting was employed to detect the manifestation of NF-κB p-p65, IL-1β, and VEGF. (**d**) Quantification of NF-κB p-p65, IL-1β, and VEGF protein expression. * *p* < 0.05, ** *p* < 0.01, and *** *p* < 0.001 vs. control group, ns indicates *p* > 0.05, # *p* < 0.05 vs. OxLDL group. (**e**) Immunofluorescent images of NF-κB p65 (red) and DAPI staining of nuclear (blue), arrows labeled NF-κB p65 expression in nuclear, bars = 50 μm. (**f**) Pearson’s correlation analysis p65 transfer to nuclear and quantified nuclear co-localization ratio, ** *p* < 0.01 vs. OxLDL group. (**g**) Western blotting was employed to detect the manifestation of NF-κB route protein in cytoplasm or nucleus with GAPDH and histone in order to maintain internal control, and quantified. Data are presented as mean ± SEM, n = 3, * *p* < 0.05, ** *p* < 0.01, and *** *p* < 0.001 vs. control group, ### *p* < 0.001 vs. OxLDL group.

**Figure 6 jcm-12-01674-f006:**
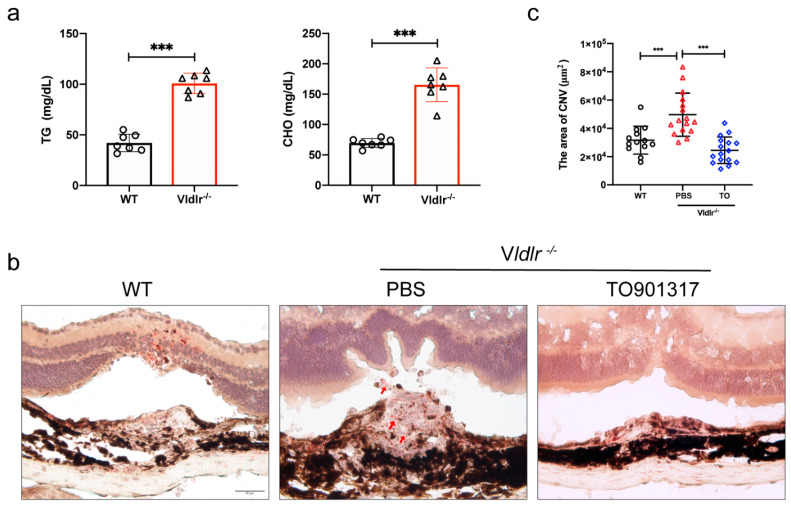
Effect of LXR agonist on *Vldlr^−/−^* mice CNV volume. (**a**) Quantification of triglycerides and total cholesterol of *Vldlr^−/−^* mice. Values are expressed as means ± SEM, n = 7, **** p* < 0.001 vs. WT group. (**b**) Oil red O staining detect lipid accumulation (red), hematoxylin labeled nuclear (blue), arrows labeled lipid accumulation, bars = 50 μm. (**c**) Quantification of CNV relative 2D lesion areas. Lesion areas were measured using ImageJ software. Values are expressed as means ± SEM, n = 15, **** p* < 0.001.

## Data Availability

The data used to support the findings of this study are included within the article.

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
