# Peer review of "Liver X Receptor Agonist Inhibits Oxidized Low-Density Lipoprotein Induced Choroidal Neovascularization via the NF-κB Signaling Pathway"

_jcm, 2023, doi:10.3390/jcm12041674_

Round 1
Reviewer 1 Report
In this interesting article, Wu et al have found that LXR agonist (TO901317) can protect the RPE cells from OxLDL mediated inflammatory and oxidative stress and CNV formation by suppressing p65 pathway. The introduction is well written and provide sufficient background for the readers to understand the results. Material method is comprehensive and sufficient enough to be reproduce by other labs.
However authors should address the concern below:
Figure.1
1. In Fig.1a, authors have used F4/80 to mark the macrophages but in the text they have mentioned as inflammatory cyotokines. They should correct it.
2. Literature suggest F4/80 is mainly expressed by the resting macrophages and its expression decreases upon macrophage activation. Therefore, Authors should also use Ly6C or Gr1 as an additional marker for staining the infiltrating inflammatory monocytes in the frozen sections of choroidal flat mount in Fig.1a-b.
3. Fig.1d is a qRT-PCR data but the text says western blot, please correct it. Also what is the concentration of TO used here? Please mention the TO concentration for other figures where its missing.
Figure.2
4. Authors need to elaborate the result shown in Fig.2b. Please explain What does the graph in Fig. 2b shows? and its significance.
5. Fig.2d, need to describe in the text which angiogenesis promoting cytokines are reduced in TO treated group. The protein array analysis should be repeated for atleast 3 times to indicate the significantly reduced cytokines.
Figure.3
6. In fig.3a, Authors should quantify the phagocytosis between the two groups and provide a bar graph with significance. The k should be in lower case in kD labels.
Figure.4
7. In unstimulated cells, p65 subunits are restricted to the cytoplasm by the inhibitory IkB (ikBalpha & beta) that binds to the p50/p65 heterodimer and masks their nuclear translocation. Upon stimulation, the IkBalpha is phosphorylated by IKK complex and degraded. Dissociation of IkBalpha induces phosphorylation of p65 and the heterodimer moves into the nucleus to induce the expression of pro-inflammatory cytokines like TNFa, IL6, IL-1b etc.
In fig.4a, authors have shown the total p65 in the immunoblot which does not reflect the true activated level of p65 as mentioned above. Authors should either use anti-phospho-p65 antibody to immunoblot total cell extract or use nuclear extract if using anti-p65 antibody. Same should be done for Fig.4c & 5a.
Figure.5
8. For fig.5a, Please use anti-phospho.p65 as for fig.4a & 4c. Also please quantify the p65 & DAPI colocalization using imageJ or any other appropriate software and provide a bar graph for fig.5c.
The RNAi data in the supplementary S1 is a crucial data, it can be transferred to the main fig.5
Figure.6
9. Please justify in few lines for the use of Vldlr-/- mice. What does vec in 6b means? Please also properly mark the lesion area in 6b with a small arrow etc.
Discussion
10. Authors indicate that “ our results demonstrates that IL-1b stimulation can activate IkBalpha .....” this is not shown in any of the data. They should remove this from the text or explain in detail how IL-1b activates IkBalpha and what is exactly meant by activation (phosphorylation, degradation etc).
Author Response
1.In Fig.1a, authors have used F4/80 to mark the macrophages but in the text they have mentioned as inflammatory cyotokines. They should correct it.
Response: Thanks for your careful revision. We felt sorry for our negligence of many writing errors in manuscript, "inflammatory cytokines" mentioned have been corrected to "F4/80 marked macrophages" and marked with colour in manuscript.
2.Literature suggest F4/80 is mainly expressed by the resting macrophages and its expression decreases upon macrophage activation. Therefore, Authors should also use Ly6C or Gr1 as an additional marker for staining the infiltrating inflammatory monocytes in the frozen sections of choroidal flat mount in Fig.1a-b.
Response: Ly6c (lymphocyte antigen 6 complex, locus C1) is a monocyte/macrophage cell differentiation antigen commonly used to differentiate classical monocytes (Ly6chigh) from non-classical ones (Ly6clow). Novel literature showed that Ly6c was expressed in the retina blood vessels [1]. Immunostaining of Ly6c in the frozen sections is additional verification for qualified monocyte/macrophage cell expression, which is added to supplement data to assist demonstrate macrophage activation.
In this study, CNV flat mount was the only quantitative method for the extent of CNV area with numerous repeats. So frozen sections were used in the supplementary Ly6c immunofluorescence staining to observe the molecular pathology within the lesion area.
- Fig.1d is a qRT-PCR data but the text says western blot, please correct it. Also what is the concentration of TO used here? Please mention the TO concentration for other figures where its missing.
Response: Thanks for your careful revision, we felt sorry for our negligence of this writing error in manuscript, and it has been corrected. All the concentrations of TO used in this research are 10μM which are added into manuscript and figure legends. For typographical reasons, the concentration of TO is written only in every figure legend.
- Authors need to elaborate the result shown in Fig.2b. Please explain What does the graph in Fig. 2b shows? and its significance.
Response: Thanks for your reminder, Fig2b is the statistical result of tube formation experiment (Fig 2a). The tube forming ability was estimated by the number of loops, the total length and branch length of tube, as reported before [2]. The aim and significance of Fig. 2b are demonstrated: TO decreases neovascularization by some angiogenic factors in vitro, which were added to the manuscript and figure legends.
- Fig.2d, need to describe in the text which angiogenesis promoting cytokines are reduced in TO treated group. The protein array analysis should be repeated for atleast 3 times to indicate the significantly reduced cytokines.
Response: According to the instructions and literature, each molecule has two duplicates in one membrane [3]. As a result, we repeat the experiment. Four replicates of cytokines were obtained now, and representative pictures were displayed. In addition, we supplemented the statistical analysis and labeled the data with statistical differences. The conclusions have been revised in the manuscript and pictures.
- In fig.3a, Authors should quantify the phagocytosis between the two groups and provide a bar graph with significance. The k should be in lower case in kD labels.
Response: The phagocytosis between two groups was quantified using the imageJ software and statistics of fluorescence intensity. The conclusions have been added to the manuscript and picture (Fig 3b). All the "KD" behind western blot bands were revised to "kD"
- In unstimulated cells, p65 subunits are restricted to the cytoplasm by the inhibitory IkB (ikBalpha & beta) that binds to the p50/p65 heterodimer and masks their nuclear translocation. Upon stimulation, the IkBalpha is phosphorylated by IKK complex and degraded. Dissociation of IkBalpha induces phosphorylation of p65 and the heterodimer moves into the nucleus to induce the expression of pro-inflammatory cytokines like TNFa, IL6, IL-1b etc.
In fig.4a, authors have shown the total p65 in the immunoblot which does not reflect the true activated level of p65 as mentioned above. Authors should either use anti-phospho-p65 antibody to immunoblot total cell extract or use nuclear extract if using anti-p65 antibody. Same should be done for Fig.4c & 5a.
Response: We are grateful for the suggestion. The western blot of p-p65 for total cell extract was detected in Fig 4a,c&5a. Statistical analysis of p-p65/p65 were added into figures and manuscript.
- For fig.5a, Please use anti-phospho.p65 as for fig.4a & 4c. Also please quantify the p65 & DAPI colocalization using imageJ or any other appropriate software and provide a bar graph for fig.5c.
The RNAi data in the supplementary S1 is a crucial data, it can be transferred to the main fig.5
Response:We are grateful for the suggestion. The western blot of p-p65 for total cell extract was repeated, and statistical analysis of p-p65/p65 in same membrane added into figures and manuscript as before. The colocalization of p65 and DAPI were quantified with imageJ software. The conclusions have been added in the manuscript and picture (Fig 5f).
- Please justify in few lines for the use of Vldlr-/- mice. What does vec in 6b means? Please also properly mark the lesion area in 6b with a small arrow etc.
Response: In our previous studies, we established an improved CNV model and added the lipid metabolism disorder into the model, making the animal model more consistent with the pathogenesis of AMD. And it was used in this research. However, we would like to further investigate animal models that can spontaneously form lipid metabolism disorders instead of exogenous injection. We found that Vldl-/- mice exhibited hyperlipidemia, as shown in result (Fig 6a). In addition, it also reduced invasive procedures.
Meanwhile, LXR helps maintain cholesterol homeostasis, not only through promotion of cholesterol efflux but also through inhibits the LDL receptor (LDLR) pathway. Vldl-/- mice is helpful for further research on the mechanism of cholesterol efflux.
“vec” in 6b means control group, which is ambiguous here. It has been corrected to “PBS” control group.
- Authors indicate that “our results demonstrates that IL-1b stimulation can activate IkBalpha .....” this is not shown in any of the data. They should remove this from the text or explain in detail how IL-1b activates IkBalpha and what is exactly meant by activation (phosphorylation, degradation etc).
Response: Previous literature studies suggested that IL-1β activation was involved in the NF-kB pathway, but there was no direct evidence in this study. Therefore, thanks for your comments and I have deleted relevant conclusions in the paper.
- Martínez-Carmona, M.;Lucas-Ruiz F.;Gallego-Ortega A.;Galindo-Romero C.;Norte-Muñoz M.;González-Riquelme M.J.;Valiente-Soriano F.J.;Vidal-Sanz M., Agudo-Barriuso M. Ly6c as a New Marker of Mouse Blood Vessels: Qualitative and Quantitative Analyses on Intact and Ischemic Retinas [J]. Int J Mol Sci, 2021, 23(1): doi: 10.3390/ijms23010019.
- Zhang, Q.;Lu S.;Li T.;Yu L.;Zhang Y.;Zeng H.;Qian X.;Bi J., Lin Y. ACE2 inhibits breast cancer angiogenesis via suppressing the VEGFa/VEGFR2/ERK pathway [J]. J Exp Clin Cancer Res, 2019, 38(1): 173. doi: 10.1186/s13046-019-1156-5.
- Ma, S.;Mangala L.S.;Hu W.;Bayaktar E.;Yokoi A.;Hu W.;Pradeep S.;Lee S.;Piehowski P.D.;Villar-Prados A.;Wu S.Y.;Mcguire M.H.;Lara O.D.;Rodriguez-Aguayo C.;Lafargue C.J.;Jennings N.B.;Rodland K.D.;Liu T.;Kundra V.;Ram P.T.;Ramakrishnan S.;Lopez-Berestein G.;Coleman R.L., Sood A.K. CD63-mediated cloaking of VEGF in small extracellular vesicles contributes to anti-VEGF therapy resistance [J]. Cell Rep, 2021, 36(7): 109549. doi: 10.1016/j.celrep.2021.109549.

Reviewer 2 Report
The main question address by the research is to evaluate the effects of Liver X Receptor on on choroidal neovascularization (CNV), associated with Age Related macular degeneration.
Topic: AMD is a topic of intense study and importance, and the manuscript content and context is relevant, and helps with futher information in the filed i.e. helps to fill a "gap" in information.
The research uses a novel method in evaluating the causes of AMD, and includes multiple factors such as lipid metabolism, cholesterol transport, and angiogenesis.
Methodology: The methodology is clear and competent, in my opinion. No further controls are deemed necessary.
Conclusions: The conclusions are consistent with the evidence and arguments presented and address the main question posed.
Tables and figures: I found the images, figures and diagrams hard to read and decipher at first until I viewed an enlarged version online. However, all seemed in order after perusing the data.
Some minor improvements in English needed for better clarity in the manuscript. while I detected 9% plagiarism, you have cited references appropriately, and the words used are acceptable. An interesting paper with sound introduction, methodology, validated results, discussion, and conclusion. Took a while to decipher all the illustrations!
Author Response
Dear Editors and Reviewers:
Thank you for your letter and the reviewers’ comments concerning our manuscript entitled “Liver X receptor agonist inhibits oxidized low-density lipoprotein” (jcm-2100620). Those comments are all valuable and very helpful for revising and improving our paper. And revised portions are marked in the paper.
Response: We appreciate the reviewer’s positive evaluation of our work. And we made some modifications to the pictures and diagrams to strengthen the evidence for the conclusion and improve the coherence of the article.